# The net effect of wealth on health—Investigating noncommunicable diseases mortality in the context of regional affluence

**Alicja Olejnik**[ID][☯], **Agata Żółtaszek**[ID][☯]*

Department of Spatial Econometrics, Faculty of Economics and Sociology, University of Lodz, Łódź, Poland

☯ These authors contributed equally to this work.
* agata.zoltaszek@uni.lodz.pl

**Data Availability Statement:** Full data set is available as a Supporting information files named: Database (Net effect).

## Abstract

The wealth-health relationship is not unambiguous and constant. Greater wealth affects individual and population health in opposite ways. Increased risk factors especially raise the probability of noncommunicable diseases (NCDs) impacting a population. Conversely, better healthcare and awareness reduce the chances of developing these diseases or increase the likelihood of treatment and cure. Therefore, this paper aims to assess and quantify the hard-to-grasp overall impact of prosperity on mortality from selected NCDs, allowing us to capture the relevant differences between European regions. In particular, we attempt to estimate the net effect of affluence and the health economic threshold of the GDP-mortality relationship, by developing a dedicated analytical tool based on joinpoint regression and forecasting methods. Our results show that in the case of most investigated diseases in more impoverished regions, a clear pattern reflects mortality rising with prosperity. After crossing the health economic threshold of around 20 thousand euros per capita, the trend changes by stabilising or reversing. The research we present shows that health policy should be more diversified locally to enable health convergence at the national and European regional levels. Moreover, health policy should evolve to prioritise mental and neurological disorders, by improving the resource allocation and increasing public awareness.

## Introduction

According to the World Health Organisation (WHO) Global Health Observatory, 71% of human deaths are due to noncommunicable diseases (NCDs), while in high-income countries, the figure can rise to 90% [1, 2]. Some well-established relationships appear between affluence and health [3–9]. Declining quality of air, water and soil, permanent mental strain, harmful eating habits, sedentary lifestyle, and exposure to artificial light are only some negative effects of modern life and socioeconomic development. These detrimental influences constitute risk factors impacting the prevalence of mental and neurological disorders, diabetes, cancer and cardiovascular and respiratory illnesses [10, 11].

In recent years, interest has grown in assessing the overall medical impact of socioeconomic conditions on NCD prevalence. Economic research evaluating the association between wealth

**Funding:** The authors received no specific funding for this work.

**Competing interests:** The authors have declared that no competing interests exist.

and health is still scarce and often contains conflicting findings [12–16]. Recent data and observations suggest that the relationship between economic development and prevalence is much more complex than expected. While harmful by-products of wealth negatively impact human health (e.g. through ubiquitous stress inherent in modern lifestyles [3], unhealthy diet, insufficient physical activity, irregular sleeping patterns [11, 17], environmental pollution [7], climate change [9]), technological development and financial resources determine the quality of healthcare and its accessibility [18]. This relationship means that more affluent locations and subpopulations have an advantage in preventing, diagnosing and treating illnesses. This advantage predominantly stems from the general population's better education, better-quality healthcare specialists and more advanced medical technology that taxes and common social or private medical insurance finance [10, 19, 20]. Therefore, the wealth-health relationship is not explicit or constant and may be heterogeneous due to the level of development.

Overall, a higher level of wealth affects the individual and the population's health in two opposite ways. Increased risk factors raise the probability of some NCDs presence, but better healthcare and awareness reduce the chances of their developing or increase the likelihood of treating and curing them [15, 16, 21]. Therefore, determining the resultant impact on health or the 'net effect' of wealth (positive or negative) may be challenging. From now on, we refer to that impact as the net effect of affluence, or net effect. Moreover, this effect may not be fixed for different income groups. Thus, it stands to reason that an affluence point changing the predominant impact of wealth (positive/negative) may exist, and we refer to it as the 'health economic threshold'.

Hence, this paper aims to assess and quantify the hard-to-grasp overall impact of prosperity on mortality from selected NCDs in the European Union (EU). In particular, we attempt to estimate the net effect of affluence and the health economic threshold using an econometric approach. Furthermore, we specify the type of net effect describing each investigated disease and verify whether capturing any spatial pattern by identifying significant differences between countries is possible.

The paper's structure is as follows. The Data section offers a description of the incorporated data and basic distributional statistics. The Methodology section presents our newly developed analytical instrument and some basics of exploratory spatial data analysis (ESDA). The Results and discussion section presents the results of the research at both regional and country levels. The final section highlights the conclusions.

## Data

To determine the overall effect of wealth on health in Europe, we include main categories of NCDs: diabetes mellitus, mental and behavioural disorders, diseases of the nervous system and the sense organs (neurological conditions), diseases of the circulatory system (cardiovascular diseases), diseases of the respiratory system and neoplasms (cancers). The regional gross domestic product (GDP) per economically active population at current market prices, by Nomenclature of Territorial Units for Statistics 2 (NUTS 2) in the purchasing power standard (PPS) (euro), proxies social, economic and technological development. Among the few available data, this indicator seems to best reflect the complex concept of regional affluence. Since age-standardised statistics describe mortality, we also incorporate the GDP per economically active population (later referred to as GDP or GDP per capita) to eliminate the population's age structure.

All the data in the subsequent analysis came from the Eurostat Database [22] with the exception of GDP for Swiss regions (obtained from [23]) and for Liechtenstein (from [24]). It covers 282 regions of the European Union, plus Norway, Switzerland, the United Kingdom

**Table 1. Basic statistics for three-year average (2013–2015), age-standardised death rates of selected NCDs (Europe, NUTS 2).**

| Disease | Mean | Std. dev. | Min | Max |
|---|---|---|---|---|
| Diabetes mellitus | 21.96 | 11.02 | 5.13 | 69.18 |
| Mental and behavioural disorders | 41.72 | 26.16 | 0.19 | 101.61 |
| Diseases of the nervous system and the sense organs | 39.36 | 19.42 | 10.52 | 162.97 |
| Diseases of the circulatory system | 402.80 | 206.87 | 162.96 | 1225.61 |
| Diseases of the respiratory system | 85.53 | 30.58 | 31.56 | 174.34 |
| Neoplasms | 269.67 | 31.67 | 210.84 | 374.51 |

(UK) and Liechtenstein. We use the mortality rate—the number of deaths by cause of death per 100 thousand inhabitants—as the prevalence statistic. Since we aim to assess affluence's final or net effect, this statistic seems most appropriate. Because NCD prevalence and mortality depend on age, the analysis is based on age-standardised death rates (three-year average to account for the heterogeneous age structure of the European population) [22, 25]. The data are 2013–2015 averages (most recent data available in September 2022); thus, we adjust the fundamental analysis to that time frame. To preserve the causal relationship between affluence (measured by GDP) and prevalence (measured by mortality rates), we allow for a time lag by using the GDP values for the primary year (2013).

Note that mortality rates are based on the information contained in the death certificates issued at the place of death, rather than in the region of residence. There is no widely available data for NCDs and NUTS 2 areas on the number of ill or diagnosed people, or NCD patients dying from other causes. Additionally, the number of deaths for regions with highly specialised clinics may be artificially inflated, distorting the analysis results in certain areas. Moreover, in some countries, data may not accurately reflect the actual situation, due to differences in laws and procedures leading to an autopsy or the lack of one. For instance, this can lead to overusing 'cardiopulmonary failure' and, consequently, artificially increasing the number of deaths attributed to cardiovascular disease.

Table 1 presents basic statistics for the included NCD mortality rates. In general, the leading causes of death appear to be cardiovascular diseases (over 400 deaths per 100 thousand of the population), and neoplasms are the second most common (almost 270 per 100 thousand of the population). However, the greatest diversity (measured by the relative standard deviation) occurs for mental and behavioural disorders (63% of the mean value). The variation is relatively low for deaths due to cancer (12%).

## Methodology

We carried out the research in two stages. First, we analysed the spatial distribution of NCD mortality rates and their spatial patterns in association with the distribution of economic development, using ESDA tools. Second, we attempted to quantify the net effect of prosperity on the mortality of selected diseases.

In latter stage, we developed a new, specially dedicated analytical tool based on an econometric approach and counterfactual analysis. We employed a joinpoint model to produce a specification that models a relationship between each disease's mortality and development levels. Moreover, this specification can exhibit nonconstant monotonicity, which might lead to assessing the change-point level of development, at which the direction or strength of the relationship in question changes. If a change in monotonicity occurs, we can designate the point at which the function breaks as a health economic threshold. The assessment of the individual patterns for each disease, together with extrapolation methods, enables calculating the

numerical values of the net effects of affluence. The calculation of this net effect occurs for the respective location and each disease. It is defined as the difference between the actual level of mortality and a hypothetical one resulting from projecting the region to a different level of affluence.

Regions do not constitute independent, isolated economies; they interact with each other. The so-called spatial weights matrix $\mathbf{W} = [w_{ij}]_{N \times N}$, where component $w_{ij}$, represents the relation between regions $i$ and $j$, can represent the mathematical description of the spatial structure. Usually, this matrix is adopted a priori and has an exogenous character. The data are spatially autocorrelated when they tend to have similar values in neighbouring regions. Spatial autocorrelation can be studied at both local and global levels. Global spatial autocorrelation means the presence of spatial dependencies on average for all regions. Local spatial autocorrelation occurs for individual region, when we observe interactions with its neighbours [26].

To investigate the regional dispersion and spatial patterns of mortality rates, we used local indicators of spatial association (LISA). The local Moran's $I_i$ statistic allows evaluating a statistically significant relationship between a value in the $i$-th region and those in neighbouring regions.

$$I_i = \frac{(z_i - \bar{z})}{\frac{1}{N} \sum_{j=1}^{N} \left(z_j - \bar{z}\right)^2} \sum_{j=1}^{N} w_{ij} \left(z_j - \bar{z}\right). \tag{1}$$

For the quantitative variable $z$, $i$ and $j$ represent regions, $\bar{z}$ describes the average of an investigated variable and $w_{ij}$ are the elements of $\mathbf{W}$. In subsequent analysis, the $\mathbf{W}$ matrix is based on a queen contiguity spatial weight matrix, first order [26–28].

When a region with a high level of the phenomenon borders regions with similarly high values, high-high clusters (hotspots) emerge. Once low values are grouped with low values, low-low clusters (coldspots) form. There are also mixed groups: low-high and high-low. We only analyse locations where local statistics significantly differed from zero.

A joinpoint regression, also known as change-point regression, can be a valuable tool for making inferences about changes in trends (see, for example, [29–33]) Among other uses, it has a role in both the analysis of time changes in suicide rates (e.g. in Denmark [34]) and the assessment of changing trends in the incidence of cancer (e.g. in Canada, Great Britain, Japan and Italy [35–38]). However, to our knowledge, no form of joinpoint regression or related model has been used to assess the change point in GDP at which the impact of affluence on the mortality rate changes its direction.

To identify and evaluate the GDP level beyond which the direction change occurs, we introduce a form of joinpoint model (2). The method splits data into two subsets, for which individual regression lines are estimated with a common intercept and two different slopes. The point that breaks the sample into two subsets is called a change point or joinpoint, as it changes the behaviour (trend) of the investigated data and simultaneously joins the two trends together. Our model is based on the functional form $y = f(x)$ and

$$f(x) = f\big(x; \alpha_0, \alpha_1, \alpha_2, \tau^*\big) := \alpha_0 - \alpha_1 (\tau^* - x)^+ + \alpha_2 (x - \tau^*)^+ + \epsilon, \tag{2}$$

Where $\alpha_0, \alpha_1, \alpha_2, \tau^*$ are regression coefficients. Parameters $\alpha_1, \alpha_2$ are slopes of two regression lines, and $\tau^*$ is a change point. Variable $x$ represents the independent variable, and the expression $(A)^+$ denotes the positive part of a number $A$. We assume that the error term ($\epsilon$) is randomly distributed with zero mean and a common bound for variance $\sigma^2$ for both subsets. This model can be estimated with an adjusted ordinary least squares method, by following the general theory of Pötscher and Prucha [39]. The resulting M-estimator turns out to be consistent

under the same assumptions as standard ordinary least squares for the regressors, and with the additional requirement of the corresponding parameter space being bounded.

In our study, function $f$ describes the mortality rate for chosen NCDs, and change point $\tau^*$ is $GDP^*$—the level at which potential change in the regression line of mortality rates occurs, the health economic threshold. Variable $GDP_i$ represents regional or national gross domestic product per capita. That is,

$$MR_i^d(GDP_i; \alpha_0, \alpha_1, \alpha_2, GDP^*) = \alpha_0 + \alpha_1(GDP^* - GDP_i)^+ + \alpha_2(GDP_i - GDP^*)^+ + \epsilon_i, \quad (3)$$

where $MR_i^d$ is the age-standardised three-year average mortality rate due to an NCD $d$ ($d = 1, \ldots, 6$). As a result, for each disease, we obtain two linear functions: $f_1$ for the subset $R_1$ of lower GDP regions (for which $GDP_i \leq GDP^*$, $i \in R_1$) and $f_2$ for the subset $R_2$ of upper GDP regions (for which $GDP_i \geq GDP^*$, $i \in R_2$), where $f_1(GDP^*) = f_2(GDP^*)$. When assessing the relationship between mortality and affluence, two aspects are relevant: tendencies (slope of the function) and levels (mortality values, especially for constant functions). Additionally, the value of $GDP^*$ and the subsequent size of subsets $R_1$ and $R_2$ are of interest.

To calculate the net effect of affluence on regional health, we use an approach based on project evaluation methodology—that is, counterfactual forecasting. These project assessment methods can extrapolate historical trends and measure the difference between the actual and the theoretical value that would be expected otherwise, without (for example) subsidy support. In project evaluation, the change point is known as it represents the starting point of the project. However, our study required estimating the change point, and we operated not on time but on cross-sectional data.

Linear functions $f_1$ and $f_2$ obtained by the joinpoint regression method:

$$MR_i^d(GDP_i; \alpha_0, \alpha_1, \alpha_2, GDP^*) = \begin{cases} f_1(GDP_i) + \epsilon_i, \text{for } i \in R_1 \\ f_2(GDP_i) + \epsilon_i, \text{for } i \in R_2 \end{cases} \quad (4)$$

can be extrapolated to the regions with GDP outside their respective domains: $R_1$ and $R_2$. That is, the theoretical extension of the function $f_2$ now covers regions with lower GDP ($R_1$), where theoretical extension of the function $f_1$ applies to the regions with higher GDP ($R_2$). Operating on those theoretical functions' extrapolations and empirical values, by calculating the difference between them, we can acquire two net effect values, namely, of poverty ($NE\_P$) and of affluence ($NE\_A$).

First, we assess how the theoretical mortality, resulting from extrapolation of the estimated function $f_2$ to the subset $R_1$, differs from the empirical mortality levels, i.e. the impact of poverty. When $MR_i^d > f_2(GDP_i)$, the actual mortality rate due to $d$-th disease in the $i$-th region is higher than that which would have been reached if the general trends, typical of areas with higher GDP, were to have emerged in this region. Therefore, the effect describes the number of people (per 100 thousand inhabitants) who died because the $i$-th region is poor, or alternatively, the number of people that could have been saved if region $i$ had been more prosperous and benefitted from better-quality healthcare. We identify this effect as a negative effect of poverty:

$$NE\_PN_i^d = \max\{MR_i^d - f_2(GDP_i), 0\}, d = 1, \ldots, 6, i \in R_1. \quad (5)$$

Adversely, $MR_i^d < f_2(GDP_i)$ means that the empirical mortality rate is lower than it would be if the region were more affluent. Hence, it describes the number of people who survived because they lived in the poorer region. This indicates the positive impact of poverty on

regional health:

$$NE\_PP_i^d = \max\{f_2(GDP_i) - MR_i^d, 0\}, d = 1, \ldots, 6, i \in R_1. \tag{6}$$

Similarly, if we extrapolate the function $f_1$ to the subset $R_2$, we obtain a theoretical function extending the trend of the more deprived regions to more affluent ones—the net effect of affluence. Comparing this extrapolation with empirical data results in the effect $NE\_A$. For $MR_i^d > f_1(GDP_i)$, we obtain the number of people who died due to living in a wealthy region (i.e. who would not have died if their regions were less privileged), describing an explicate (true) negative effect of affluence:

$$NE\_AN_i^d = \max\{MR_i^d - f_1(GDP_i), 0\}, d = 1, \ldots, 6, i \in R_2. \tag{7}$$

In the opposite case, $MR_i^d < f_1(GDP_i)$, we get the number of people who survived thanks to the region's prosperity. We recognise this as a direct positive effect of affluence:

$$NE\_AP_i^d = \max\{f_1(GDP_i) - MR_i^d, 0\}, d = 1, \ldots, 6, i \in R_2. \tag{8}$$

The proposed model and methodology are designed to detect the overall tendency, not to estimate the direct impact of GDP on mortality. This study should appear as an extensive correlation analysis, not as a cause-and-effect model. To obtain the latter requires controlling for other critical socioeconomic factors, among others, the level of education and healthcare systems. In our study, controlling for other determinants would come at the cost of a correct economic interpretation of the economic threshold, which requires using a single criteria setting.

## Results and discussion

In the first part of the study, we examined the regional dispersion and spatial patterns of mortality rates in Europe as well as the distribution of GDP per capita. Żółtaszek and Olejnik [40] carried out a similar, though more detailed, analysis of death rates in 2003–2005 and 2008–2010. Figs 1 and 2 show the spatial distribution and LISA analysis for all standardised death rates of investigated NCDs.

Cancer mortality is highest in Poland, Slovakia, Hungary, Croatia, Denmark, the northern UK, parts of the Netherlands and the Czech Republic. Regions with low mortality are mainly in southern Europe (excluding Croatia) and Scandinavia.

Cardiovascular diseases have an apparent spatial pattern. Regions with high mortality rates are in Central and Eastern Europe (CEE), particularly in the Balkans and the Baltic States. France, Spain and the southern UK have the lowest death rates. The risk of developing a cardiovascular disease varies by sex (higher for males) and increases with age, smoking, cholesterol levels and body mass index, as WHO indicated. Notably, the corresponding prevalence probabilities (by the category of risk factors) are higher for CEE than for Western Europe (WE) [41–43]. The cluster map (Fig 2) reflects this bipolar East-West division.

In cases of mortality that diabetes caused, the highest values in 2013–2015 appeared in Portugal, Germany, Hungary, Austria, the Czech Republic, southern Italy, Cyprus, Croatia and a few regions in Poland, as well as the Severoiztochen region in Bulgaria. Low mortality occurred mainly in the UK, Finland and Greece. Significant hotspots and coldspots across Europe reflected this distribution. Globally, the prevalence of diabetes has been rising more rapidly in low- and middle-income countries than in high-income ones. The key determinants include unhealthy diet, regular physical activity, body weight and tobacco [44].

A different tendency occurs for mental and behavioural disorders and neurological diseases. The former account for the highest number of deaths in the UK, Scandinavia, the

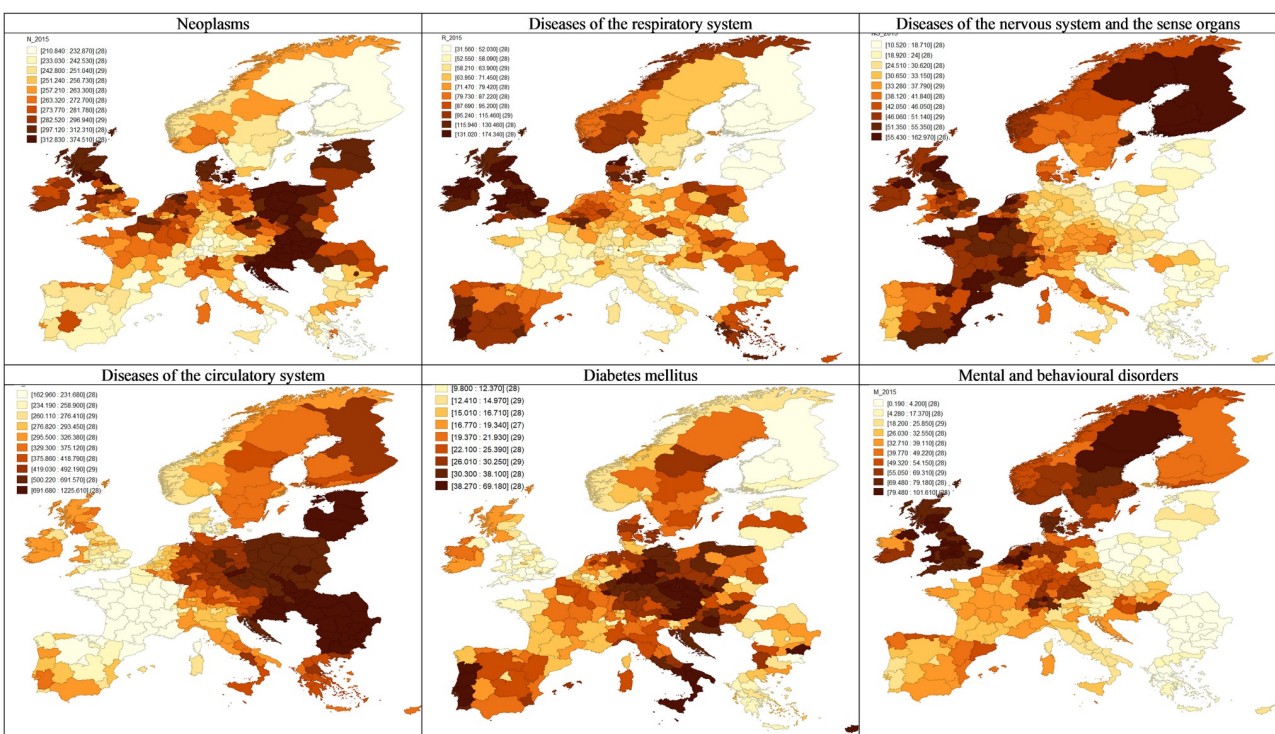

**Fig 1. Standardised death rates of diseases of affluence, average 2013–2015 by region of residence, in deciles.** (Own compilation in GeoDa software, based on Eurostat data and shapefile).

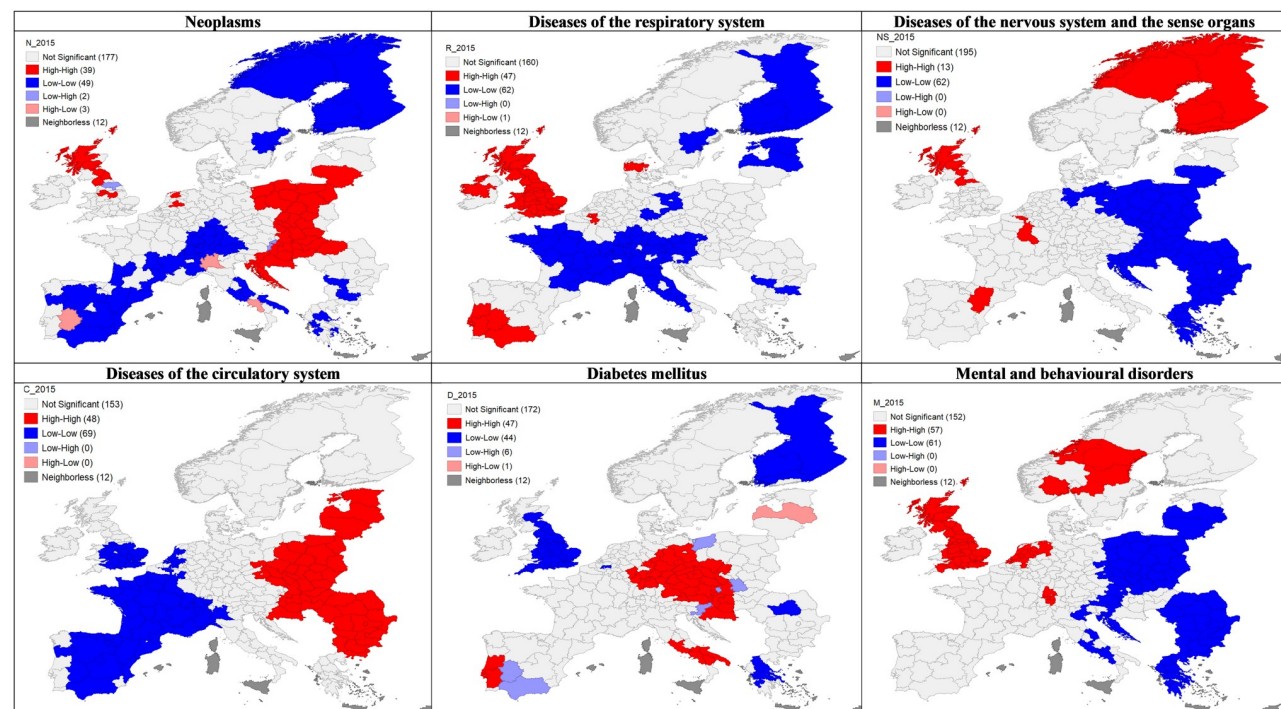

**Fig 2. Univariate LISA for diseases of affluence standardised death rates, averages 2013–2015 by NUTS 2 EU regions.** (Own compilation in GeoDa software, based on Eurostat data and shapefile).

Netherlands and Switzerland, creating four hotspot clusters. In turn, low mortality occurs in CEE, except for Hungary, nearly forming one large cluster. Nervous system and sensory diseases have the highest mortality rates in Scandinavia (especially Finland), with rates, on average, three times higher than in other countries, reflecting a distinct hotspot. Regions with relatively high mortality are in France, southern Spain, Benelux, Great Britain and Ireland. CEE has minor death rates due to diseases of the nervous system, forming a voluminous coldspot. For both groups of disorders, we observe a similar spatial pattern, opposite to that of cardiovascular conditions. Notably, mental illnesses are the leading cause worldwide of years lived with disability and premature death. Females, children, adolescents and people with traumatic life experiences constitute the core of the vulnerable population [45].

Respiratory diseases are the most common cause of death in the UK, Ireland, the exterior Iberian Peninsula, Denmark and parts of Greece, Belgium and Norway. The southern part of the Iberian Peninsula and Great Britain, except for Northern Ireland, are the main mortality hotspots. The lowest mortality rates appear in Finland, the Baltic states, France, Switzerland, Austria, Croatia and part of Italy, which the large coldspots on the LISA map confirm.

To analyse the relationship between mortality rates and affluence, we consider respective Pearson's correlation coefficients (Table 2) together with the spatial distribution of GDP (including clusters) in Fig 3. Among selected NCDs, mental and behavioural disorders have the strongest positive and significant relationship with affluence. More impoverished regions of Poland and the Balkans have much lower mortality than more prosperous provinces in the UK, Scandinavia and Benelux. A similar pattern appears for diseases of the nervous system and the sense organs. These two categories of illness seem to reflect the negative impact of affluence. In turn, diabetes, cardiovascular conditions and neoplasms show opposite tendencies, with a negative but significant correlation to GDP. More affluent regions that correspond with a lower mortality rate may indicate better healthcare. In the case of respiratory system diseases, there is no linear correlation between GDP and mortality.

Subsequently, we assessed the net effect of affluence on mortality and health economic threshold. Table 3 presents the estimation results of joinpoint regression that the Methodology describes. The analysis shows that the health economic threshold $GDP^*$ differs for each disease group. However, except for neoplasms, all change points are around 20 thousand euros per capita, ranging from 17 thousand for diabetes to almost 23 thousand for mental and behavioural disorders. As a result, for each disease category, the sets of more impoverished ($R_1$) and more affluent ($R_2$) regions are very similar. Thus, 20 thousand euros is the limit at which the positive or negative wealth impact trend changes or reverses. Fig 4 offers the plots of joinpoint regression for each disease, based on the estimation results (cf. formula 3); Fig 5 presents the spatial distribution of calculated net effects (cf. formulas 5–8).

**Table 2. Correlation coefficient with GDP per inhabitant, 2013, with average selected age-standardised mortality rates, 2013–2015.**

| Disease | Correlation coefficient | p-value |
|---|---|---|
| Neoplasms | −0.222*** | 0.0002 |
| Diseases of the circulatory system | −0.393*** | 0.0001 |
| Diabetes mellitus | −0.125** | 0.0360 |
| Mental and behavioural disorders | 0.308*** | 0.0001 |
| Diseases of the nervous system and the sense organs | 0.235*** | 0.0001 |
| Diseases of the respiratory system | −0.058 | 0.3310 |

(Significance level: *−10%, **– 5%, ***– 1%)

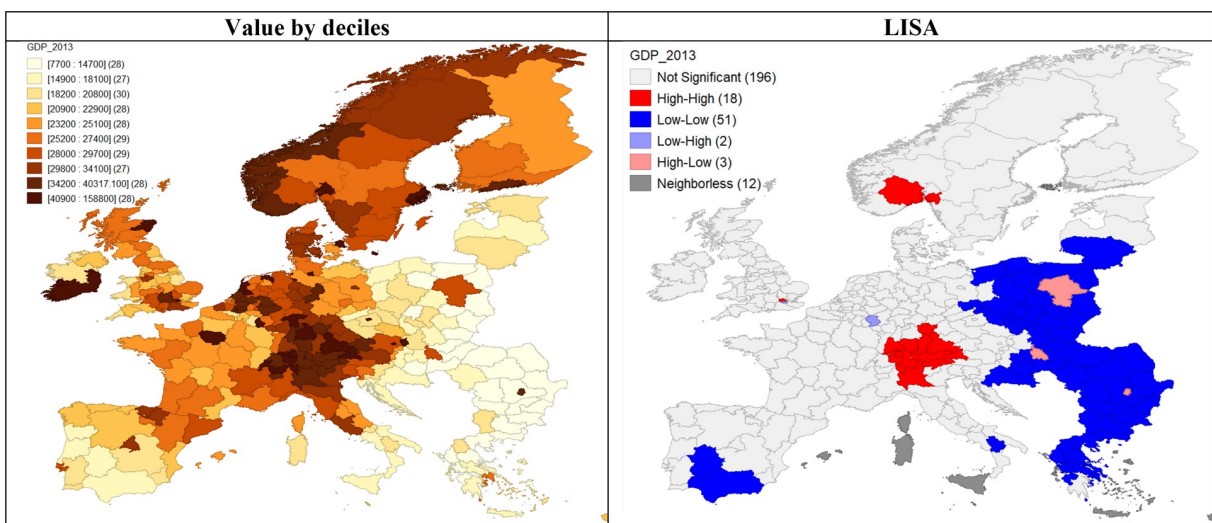

**Fig 3. GDP in euro per inhabitant, 2013, by NUTS 2 EU regions.** (Own compilation in GeoDa software, based on Eurostat data and shapefile).

**Table 3. Results of joinpoint regression with an estimated health economic threshold—*GDP\** for selected NCDs age-standardised mortality rates, 2013–2015.**

| Disease | Parameter | | t-stat | GDP* |
|---|---|---|---|---|
| **Neoplasms** | | | | 37 286.59 |
| | $\alpha_1$ | −0.0011*** | 4.71 | |
| | $\alpha_0$ | 256.63*** | 77.37 | |
| | $\alpha_2$ | – | – | |
| **Diseases of the circulatory system** | | | | 21 397.82 |
| | $\alpha_1$ | −0.0514*** | 19.01 | |
| | $\alpha_0$ | 321.09*** | 34.77 | |
| | $\alpha_2$ | – | – | |
| **Diabetes mellitus** | | | | 16 999.90 |
| | $\alpha_1$ | – | – | |
| | $\alpha_0$ | 23.21*** | 27.19 | |
| | $\alpha_2$ | −0.0001** | 2.28 | |
| **Mental and behavioural disorders** | | | | 22 893.41 |
| | $\alpha_1$ | 0.0041*** | 11.20 | |
| | $\alpha_0$ | 50.42*** | 33.29 | |
| | $\alpha_2$ | – | – | |
| **Diseases of the nervous system** | | | | 22 301.10 |
| | $\alpha_1$ | 0.0028*** | 9.04 | |
| | $\alpha_0$ | 44.60*** | 37.93 | |
| | $\alpha_2$ | – | – | |
| **Diseases of the respiratory system** | | | | 17 710.68 |
| | $\alpha_1$ | 0.0026** | 2.43 | |
| | $\alpha_0$ | 90.25*** | 34.85 | |
| | $\alpha_2$ | −0.0003** | 1.98 | |

(Significance level: *– 10%, **– 5%, ***– 1%)

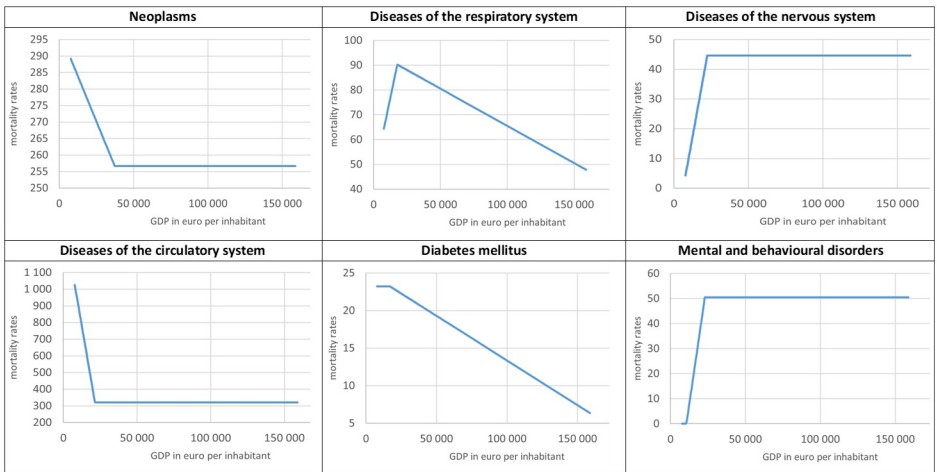

**Fig 4. Plot for joinpoint regression with an estimated change point for selected NCDs age-standardised mortality rates, 2013–2015.** (Own compilation MS Excel, based on own calculations).

In the case of neoplasms and diseases of the circulatory system, the joinpoint regression function patterns seem similar. With growing wealth, the mortality decreases (for subset $R_1$) and stabilises after passing the health economic threshold. That means that no matter how wealthy the region is, reducing the number of deaths above the health economic threshold is impossible.

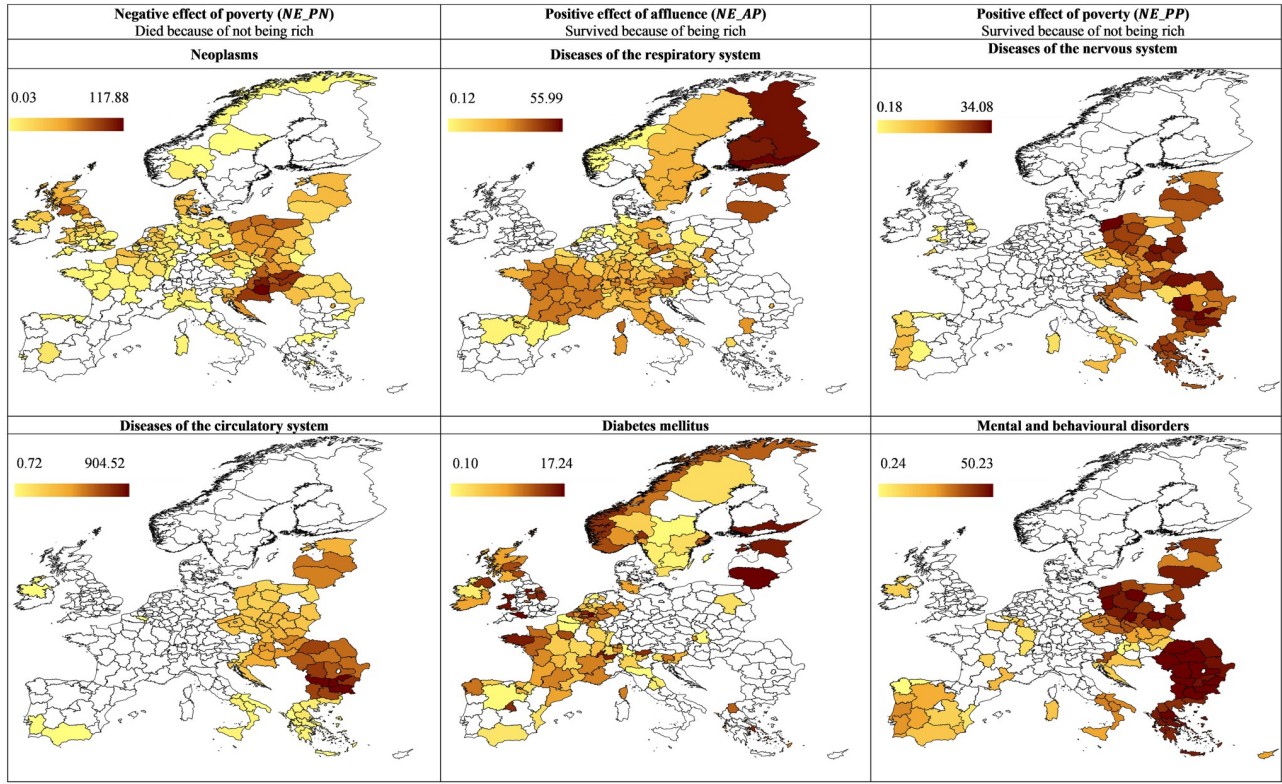

**Fig 5. Absolute values of the net effect of GDP for selected NCDs age-standardised mortality rates, 2013–2015.** (Own compilation in ArcMap software, based on own calculations and Eurostat shapefile).

Contrary to its incidence, the cancer death rate is higher in poorer regions than in more affluent ones, due to the lower level of effectiveness of oncological treatment. For instance, the number of new cases of breast cancer (per 100 thousand women) is greater in WE, while the mortality rate is noticeably higher in CEE. The European Cancer Information System (ECIS) reports that the cause of a higher number of new cases is not only greater prevalence (e.g. due to 'reproductive factors, increasing obesity and physical inactivity') but also screening intensity. ECIS also emphasises that mortality due to breast cancer declines primarily because of 'effective treatment and tools for detecting the disease at early stages' [46]. A similar pattern occurs for colorectal cancer, resulting from uneven healthcare expenditures and the quality of screening, diagnosis and treatment [47]. Also, the WHO highlights that the critical barriers to the early diagnosis and treatment of cancer include financial and logistic obstacles—that is, availability and affordability—that disadvantage low-income countries. Furthermore, delaying the treatment makes it more expensive, less effective and disabling, which burdens less affluent regions [19, 48]. Our study confirms these observations as the mortality is greater in more impoverished regions.

The estimated health economic threshold for neoplasms is the highest among the selected diseases (37 thousand euros per capita). Hence, a stabilisation of the mortality level appears only in the 38 wealthiest regions in Europe, and a further reduction in cancer mortality does not appear to be possible. This confirms the WHO findings that some types of neoplasms have less or decreasing (over time) mortality only in countries with the highest Human Development Index than those with low, medium and even high levels of development. This suggests that the predetermined number of deaths (in Europe, 257 people per 100 thousand inhabitants) depends not on the system solutions but on external determinants. Unwillingness or inability to avoid the principal risk factors and psychological, social or cultural barriers to consulting health professionals may be inevitable costs of socioeconomic development [19]. Hence, the levelling of mortality for more prosperous regions can appear as both a positive (because of relatively low mortality) and a negative (no further decrease in mortality with higher GDP is possible) effect of affluence.

On the other hand, 87% of NUTS 2 regions reside in the more impoverished subset. While the descending tendency is a benefit of increasing prosperity, soaring death rates burden some areas. The net effect map shows the distribution of people who died due to the region's low prosperity (negative effect of poverty—*NE_PN*; see Fig 2). The highest mortality and the most significant loss from economic underdevelopment were recorded in Southern Transdanubia (HU23), in the southwestern part of Hungary. In 2013, the area was the 12th least-developed European region [49]. The former coal and uranium mining region, sparsely populated with many poorly accessible settlements, has an almost 39% higher cancer mortality rate than the European average (374.51 HU23 vs 269.28 EU). The estimated excessive number of deaths for this region is 117.88.

Generally, CEE states are encumbered with excessive mortality. Due to a high economic threshold ($GDP^* \approx 37$ thousand euros per capita), even many northern parts of Germany and France, as well as Denmark, Benelux and the UK, face a similar problem. The results highlight the importance of affluence in reducing mortality through cancer screening, early detection and treatment, conditional upon availability and affordability.

With a slope of −0.05, the cardiovascular mortality rate shows a steep decline with increasing affluence for 32% of less prosperous regions, suggesting a positive effect of wealth. On the other hand, the absolute number of additional deaths, which net effect (negative effect of poverty, i.e. *NE_PN*) describes, is exceptionally high in CEE, up to 905 deaths (per 100 thousand inhabitants) in Northwestern Bulgaria (BG31). With the lowest-ranked economy in Bulgaria and the EU as a whole, this region has the lowest life expectancy in Europe. Bulgaria's average

death rate from coronary heart disease is 1,000 compared to almost 400 for the EU [22]. In 2015, the cumulative number of persons living with cardiovascular disease (age-standardised per 100 thousand) in Bulgaria was 44% higher than the EU average [50].

The bipolar East-West net effect pattern that emerged from our study corresponds with the WHO cardiovascular disease risk assessment for Europe. According to WHO, in all subcategories (e.g. age, sex, diabetes, smoking), the chances of acquiring a coronary issue are noticeably higher in CEE than in Western countries. Globally, over 75% of deaths due to cardiovascular problems occur in low- and middle-income countries [51, 52].

After passing the health economic threshold of 21 thousand euros per capita, the cardiovascular death rate stabilises at 321 deaths (per 100 thousand inhabitants)—a quarter of the maximal value. However, it is still higher than for any other disease, except for 20 regions in which neoplasms rank higher. Similar to cancer, after reaching $GDP^*$, a further reduction in mortality seems impossible—again, illustrating simultaneously the positive and negative wealth effects. This may indicate that a healthy and active life does not reach a particular group of people, preventing even a well-financed and well-equipped healthcare system from protecting them. As WHO suggested, addressing certain behavioural risk factors (tobacco use, harmful drinking patterns, unhealthy diet, obesity and physical inactivity) can avert most cardiovascular problems. Therefore, the reluctance to change lifestyle and undergo regular diagnostics may create an impassable boundary to preventing the further decline in mortality accompanying the increase in wealth [52].

The lowest health economic threshold of 17 thousand euros per capita is for diabetes. In the case of 16% of the poorest regions, the estimated death rate is around 23 per 100 thousand inhabitants. All regions more affluent than the change point experience a positive effect of prosperity, namely, the decreasing mortality function. As a result, the $NE\_AP$ describes the positive impact of wealth; the actual death rates are lower than those extrapolated from the tendency of more impoverished regions. The highest net effect appears in France, Norway, parts of the UK, Lithuania and Estonia. The most significant number of spared lives due to affluence (17 people per 100 thousand inhabitants) characterises București-Ilfov (RO32), which includes Romania's capital city and ranks 54th among the most prosperous regions in Europe.

As with previously mentioned diseases, early diagnosis and treatment are vital to preventing premature deaths from diabetes. However, the leading pharmaceuticals include basic diagnostics, such as blood glucose testing, which should be—and, in EU states, often are—available in primary healthcare, along with affordable insulin. Compared to expensive procedures and medications that treating other NCDs requires, reducing diabetes mortality is relatively inexpensive [17, 53]. Screening, prevention and awareness of a healthy lifestyle, rather than pricy advanced treatment, determine the positive impact of wealth. This may also be the reason for the low health economic threshold. Additionally, preventing deaths due to diabetes seems to have no limit, enhancing affluence's beneficial effect.

Diseases of the nervous system and the sense organs often link or overlap with mental and behavioural disorders [54]. Hence, it is no surprise that the health economic threshold is almost identical for both categories (just over 22 thousand euros per inhabitant), and they exhibit a similar pattern. For more impoverished regions, the higher the GDP is, the higher the mortality rate becomes, with a slope parameter equal to 0.004 for mental disorders and 0.003 for nervous system diseases. After crossing the health economic threshold, more affluent areas, utilising well-functioning and well-equipped health services, balance the consequences of those diseases and stop that disturbing trend.

Unfortunately, for more prosperous regions, no trend reversal appears. Mortality stabilises at 45 people per 100 thousand population for nervous system diseases and 50 for mental and behavioural disorders. This pattern may be perceived as a positive effect of poverty; the poorer

the region is, the lower is the mortality (*NE_PP*). Limiting the increasing mortality trend is only a semipositive effect of affluence, i.e. no further rise for wealthier regions. This is partly a consequence of the growing prevalence that accompanies affluence. Current healthcare systems find it challenging to overcome the stress, sleep deprivation and unhealthy diet habits of modern lifestyles.

The number of people who did not die from mental and neurological disorders due to living in impoverished regions is the greatest in CEE, the Iberian Peninsula and southern Italy. The top values occur in Poland, for mental disorders in the Opolskie (PL52, 50 people per 100 thousand inhabitants) and neurological conditions in the Zachodniopomorskie (PL42, 34 people per 100 thousand population). It seems that less prosperous regions may benefit from lower prevalence.

Numerous neurological and mental conditions can be congenital or hereditary or onset due to injury or illness. Still, risk factors (e.g. lifestyle, stress, older age, drug and alcohol use and environmental exposure, such as pollution) are also of great importance [54, 55]. Moreover, though effectively treating those diseases incurs a relatively low cost, according to WHO, in most countries, the quality and quantity of healthcare are insufficient to meet the demand. Even prosperous countries do not adjust the structure of medical services to the evolving prevalence of neurological and mental disorders. This disparity between the need and the supply results from 'severe human rights violations, discrimination, and stigma' and obsolete health policies [56, 57].

In the case of respiratory system diseases, the correlation coefficient with GDP is insignificant. However, the correlation for the subset of more impoverished regions is positive (0.3, $p = 0.03$), while for the subset beyond the threshold, it is negative ($-0.12$, $p = 0.07$) and both qualify as significant. Nevertheless, interpreting the estimation results must proceed cautiously and draw no definitive conclusions. We may infer that probably for poorer regions, mortality increases with higher GDP, indicating a disease of affluence. Above the health economic threshold of almost 18 thousand euros per inhabitant, the function changes, and higher prosperity corresponds with lower death rates. The net effect *NE_AP* (positive effect of affluence) reflects the number of people that survived because they lived in an affluent region. The highest values appear in Finland, with a maximum of 56 people per 100 thousand inhabitants for South Finland (FI1C), Sweden and Western countries.

As the WHO studies indicate, risk factors are critical determinants of health in many respiratory illnesses. The most common include smoking, air pollution, occupational chemicals and dust, obesity and frequent lower-respiratory infections during childhood [58]. Asthma, the most common respiratory and chronic disease among children, burdens low- and lower-middle-income countries with the highest mortality [59].

To underpin the observed bipolarisation of Europe, we analyse the country's average values of net effects for each disease (Table 4).

As expected, both positive and negative impacts of poverty dominate in CEE. For circulatory system diseases and neoplasms, we assess the negative effect of poverty. In each CEE country, on average, an additional 400 people (per 100 thousand inhabitants) died because of cardiovascular diseases, from 85 in Slovenia to 815 in Bulgaria. For neoplasms, 46 people (per 100 thousand inhabitants) died across CEE, from 1 in Bulgaria to 100 in Hungary. For both diseases, the dispersion among the CEE states is moderate. We do not observe additional deaths due to these two diseases in the WE countries. Some outliers occur, due to circulatory problems in Greece (66 deaths) and neoplasms in Denmark (39), the Netherlands (28) and the UK (25).

A positive effect of poverty characterises neurological and mental disorders. Each CEE state's additional average surviving rate is 20 and 30, respectively. The greatest value for

**Table 4. Country net effects for CEE and WE states by disease.**

| | Country | Diseases of the circulatory system (NE_PN) | Neoplasms (NE_PN) | Diseases of the nervous system (NE_PP) | Mental and behavioural disorders (NE_PP) | Diabetes mellitus (NE_AP) | Diseases of the respiratory system (NE_AP) |
|---|---|---|---|---|---|---|---|
| CEE | Bulgaria | 815 | 1 | 29 | 49 | 0 | 5 |
| | Czechia | 294 | 31 | 12 | 29 | 0 | 8 |
| | Estonia | 373 | 46 | 21 | 41 | 11 | 46 |
| | Croatia | 353 | 79 | 21 | 15 | 0 | 0 |
| | Hungary | 419 | 100 | 20 | 5 | 0 | 3 |
| | Lithuania | 551 | 26 | 23 | 46 | 12 | 42 |
| | Latvia | 569 | 46 | 27 | 30 | 0 | 0 |
| | Poland | 271 | 53 | 24 | 42 | 0 | 3 |
| | Romania | 572 | 23 | 20 | 42 | 2 | 2 |
| | Slovakia | 275 | 50 | 12 | 17 | 1 | 3 |
| | Slovenia | 85 | 51 | 12 | 18 | 6 | 17 |
| | **Mean** | **416** | **46** | **20** | **30** | **3** | **12** |
| | **CV** | **46%** | **55%** | **28%** | **45%** | **145%** | **133%** |
| WE | Austria | 0 | 0 | 0 | 0 | 0 | 35 |
| | Belgium | 1 | 12 | 0 | 4 | 8 | 0 |
| | Switzerland | 0 | 0 | 0 | 0 | 7 | 27 |
| | Cyprus | 0 | 0 | 0 | 26 | 0 | 0 |
| | Germany | 0 | 7 | 0 | 0 | 1 | 14 |
| | Denmark | 0 | 39 | 0 | 0 | 0 | 0 |
| | Greece | 66 | 1 | 23 | 42 | 2 | 1 |
| | Spain | 2 | 2 | 0 | 7 | 3 | 2 |
| | Finland | 0 | 0 | 0 | 0 | 14 | 43 |
| | France | 0 | 8 | 0 | 2 | 5 | 30 |
| | Ireland | 2 | 12 | 0 | 6 | 3 | 0 |
| | Italy | 15 | 4 | 3 | 9 | 1 | 20 |
| | Liechtenstein | 0 | 0 | 0 | 0 | 1 | 0 |
| | Luxembourg | 0 | 0 | 0 | 0 | 1 | 2 |
| | Malta | 0 | 0 | 0 | 0 | 0 | 0 |
| | Netherlands | 0 | 28 | 0 | 0 | 3 | 4 |
| | Norway | 0 | 2 | 0 | 0 | 8 | 0 |
| | Portugal | 1 | 0 | 10 | 20 | 0 | 0 |
| | Sweden | 0 | 0 | 0 | 0 | 2 | 22 |
| | United Kingdom | 0 | 25 | 0 | 0 | 12 | 0 |
| | **Mean** | **4** | **7** | **1** | **5** | **3** | **10** |
| | **CV** | **326%** | **149%** | **281%** | **183%** | **104%** | **136%** |

CV- coefficient of variation, CEE–Central and Eastern Europe, WE–Western Europe.

nervous system diseases is in Bulgaria (29) and Latvia (27), while for mental and behavioural disorders, it occurs in Bulgaria (49), Lithuania (46), Poland and Romania (42). For diseases of the nervous system, the net effect has the most even distribution across the selected diseases (coefficient of variation of 28%). In WE, the exceptions are Greece and Portugal for both diseases and Cyprus for mental disorders.

The positive effect of affluence appears for diabetes and diseases of the respiratory system. Thus, predictably, they are present in richer regions of WE; yet, values across the CEE and WE

states are very low and similar, with high dispersion within each group. The lowest values of health economic thresholds might explain this. The highest number of people who survived because they lived in richer regions were in Estonia and Lithuania for CEE and Finland and Austria for WE.

## Conclusions

A predetermined list of Western diseases has never been compiled because the relationship between health and wealth is complex and impermanent in time and space. The same ailment can be a disease of affluence in one part of the world and aligned with poverty in another. With the advancements in medicine and technology, which also link to affluence level, eliminating or at least treating some conditions can make the negative wealth factor irrelevant. At the same time, social, economic and technological development constantly introduces new risk factors. Therefore, the relationship between wealth and 21st-century diseases is not linear or constant in strength and sign. Hence, this relationship is elusive and often indirect, making distinguishing, analysing and counteracting it difficult.

Our paper assessed and quantified the net effect of affluence on the prevalence of selected noncommunicable diseases in Europe. We demonstrated that the link between those diseases and wealth undoubtedly exists, and in most cases, this correlation is nonnegative for more impoverished regions. Hence, for less affluent locations, prosperity increases mortality. A clear CEE vs. WE health division for almost all investigated diseases reflects this. The positive effect of affluence, demonstrated in the case of respiratory diseases and diabetes, characterises WE countries. The positive (for neurological and mental disorders) and negative (for cancer and cardiovascular illnesses) impact of poverty occurs mainly in CEE.

However, our study shows that for each disease, a certain level of wealth reverses or blocks this disturbing trend. Overall, the health economic threshold is around 20 thousand euros, with the most challenging value reaching almost twice as high for cancer. For diseases of the circulatory system and neoplasms, we observe that for poorer regions, the greater the prosperity is, the less is the mortality. After crossing the health economic threshold, a stabilisation occurs. The overall effect of affluence is positive for these conditions, as well as for diabetes and respiratory conditions.

Conversely, for neurological diseases and mental and behavioural disorders, the global impact of wealth on health is detrimental. Among more impoverished regions, the mortality is higher for those more prosperous, and, the mortality in wealthier regions stabilises at a disturbingly high level. Interestingly, the lowest estimated health economic thresholds (around 17 thousand euros per inhabitant) occur in the context of the positive effect of affluence in WE countries.

Notably, while the positive and negative effects of poverty appear in all CEE states, in WE, Greece and Portugal show similar characteristics. The positive effect of affluence for respiratory system diseases and diabetes is equally present in CEE and WE countries, due to the lower health economic threshold.

None of the investigated diseases seemed to exhibit the true negative effect of wealth (*NE_AN*)—that is, an unbounded rise of mortality with wealth in affluent regions. This finding does not negate the harmful impact of affluence on the prevalence of diseases and death rates. Nonetheless, it emphasises the role of well-functioning healthcare systems and enhanced preventive actions combined with higher levels of awareness. Note that for five NCDs, health economic thresholds are relatively low (below 23 thousand euros per inhabitant). Since just a fraction of regions are disadvantaged, it stands to reason that healthcare policy should be more locally diversified to allow for national and European health convergence. A more localised

perspective and approach would counteract the adverse effects of wealth and improve the population's health. The exception is neoplasms, for which the expensive treatments exceed most healthcare systems' financial and technical capabilities in most regions. This conclusion—i.e. cancer treatment is often not affordable—aligns with WHO general findings [19].

Both our results and other studies [45, 60] prove that due to the increasing prevalence and mortality, neurological diseases and mental and behavioural disorders are the plague of the 21st century. Unfortunately, no positive effect of affluence is evident (i.e. no level of GDP would reverse the trend), so a major change in healthcare approach seems necessary. Evolving health policy should prioritise these medical conditions, diverting resources and increasing public awareness. Only directly addressing this issue with a long-term national policy, including more efforts towards prevention, diagnosis and early treatment, can counteract the hazardous consequences of prosperity.

Other research explicitly addressing the wealth-health relationship allows for a broader view of the issue. Semyonov, Lewin-Epstein and Maskileyson [61] showed that on the country level, the population's general health tends to increase with the level of economic development, and in microscale, people with higher-level socioeconomic status are healthier than the poorer population. However, the study strongly indicated that economic inequality adversely impacts health, and the relation between wealth and its disparity with health may vary across subpopulations. Babones [62] also found evidence of the link between income inequality and population health. While our research did not include individual data, the assessment occurred on a subnational level. Thus, our results confirm that in fact, regional inequality of affluence does differentiate population health as well as the strength and direction of the wealth-health relation. Additionally, some studies highlight the possibility that the relation between health and wealth (or the economy) may be two-way [63, 64] and partially indirect [65], which may increase the ambiguity of the wealth-health interaction. This study did not address these angles explicitly.

The introduction of a joinpoint regression combined with the concept of a net effect to assess mortality patterns has shown promising results. We believe that our methodology may help to solve other health economics issues, with ample room for its use evident. For instance, a similar study could be performed on other diseases at a different aggregation level. It would also be worth combining outcomes of prevalence and mortality.

## Supporting information

**S1 Data.**
(XLSX)

## Author Contributions

**Conceptualization:** Alicja Olejnik, Agata Żółtaszek.

**Formal analysis:** Alicja Olejnik, Agata Żółtaszek.

**Investigation:** Alicja Olejnik, Agata Żółtaszek.

**Methodology:** Alicja Olejnik, Agata Żółtaszek.

**Visualization:** Alicja Olejnik, Agata Żółtaszek.

**Writing – original draft:** Alicja Olejnik, Agata Żółtaszek.

**Writing – review & editing:** Alicja Olejnik, Agata Żółtaszek.

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
