## [Decision Letter · Decision Letter 0]

15 Sep 2023

PONE-D-23-17022The Net Effect of Wealth on Health for Non-Communicable DiseasesPLOS ONE

Dear Dr. Żółtaszek,

Thank you for submitting your manuscript to PLOS ONE. After careful consideration, we feel that it has merit but does not fully meet PLOS ONE’s publication criteria as it currently stands. Therefore, we invite you to submit a revised version of the manuscript that addresses the points raised during the review process.

ACADEMIC EDITOR:  Please see the recommendations of the reviewers, and take them into account.

We look forward to receiving your revised manuscript.

Kind regards,

Valentina Diana Rusu, PhD

Academic Editor

PLOS ONE

Journal Requirements:

3. We note that Figures 1-3 and 5 in your submission contain [map/satellite] images which may be copyrighted. All PLOS content is published under the Creative Commons Attribution License (CC BY 4.0), which means that the manuscript, images, and Supporting Information files will be freely available online, and any third party is permitted to access, download, copy, distribute, and use these materials in any way, even commercially, with proper attribution. For these reasons, we cannot publish previously copyrighted maps or satellite images created using proprietary data, such as Google software (Google Maps, Street View, and Earth). For more information, see our copyright guidelines: http://journals.plos.org/plosone/s/licenses-and-copyright.

a. You may seek permission from the original copyright holder of Figures 1-3 and 5 to publish the content specifically under the CC BY 4.0 license. 

Reviewers' comments:

Reviewer's Responses to Questions

**Comments to the Author**

1. Is the manuscript technically sound, and do the data support the conclusions?

Reviewer #1: Yes

Reviewer #2: Yes

2. Has the statistical analysis been performed appropriately and rigorously? 

Reviewer #1: Yes

Reviewer #2: Yes

3. Have the authors made all data underlying the findings in their manuscript fully available?

Reviewer #1: Yes

Reviewer #2: Yes

4. Is the manuscript presented in an intelligible fashion and written in standard English?

Reviewer #1: No

Reviewer #2: Yes

5. Review Comments to the Author

Reviewer #1: Thanks for the opportunity to review this manuscript. This article presents a valuable and thorough methodology to assess the impact of wealth on health in Europe. The author presents a comprehensive dataset from the Eurostat Database as well as an in-depth and robust methodology founded on Exploratory Spatial Data Analysis, joinpoint regression and counterfactual analysis.

The introduction and data collection provided are well-structured and adequately discussed. The materials used for the analysis are explained and rationalised, and the author acknowledges the potential challenges of analysing data on mortality.

Specific comments:

- Intro: “This advantage predominantly stems from better education of the general population, higher quality of healthcare specialists, more advanced medical technology financed through general taxes and common social or private medical insurance”. This statement needs to be backed by evidence/reference.

- Intro: “Overall, a higher level of wealth affects the individual and the population's health in two opposite ways”. Again, more evidence/ref is required here on how these assumptions/suggestions are made.

- Method: very good to see the authors considered the issue of mortality data being standardized for age, and accordingly, they opted to use GDP data for only people in age productive groups. It is also appropriate that they allow for a time lag. Overall, the methodology to evaluate the regional dispersion of mortality rates and spatial patterns is demonstrated well. The author presents a novel approach using a joinpoint model to model the relationship between mortality and development, and the importance of changes in monotonicity is discussed. An appropriate statistical method - Local Moran’s statistic - is used to evaluate the data, and the method is properly explained. The use of counterfactual forecasting is also valuable.

- Results and discussions: the findings are well presented. One area of development could be to enrich the discussion further with contextualising external determinants of health disparities and providing some causal explanations. For example, the authors state that some types of neoplasms have lower or decreasing mortality rates in countries with higher Human Development Index, however further analysis of this connection would add further insight into the methodology and results.

- Conclusion: the whole section lack any comparison with similar literature to situate the paper’s finding in the literature. Some of the papers that came to mind are:

https://www.sciencedirect.com/science/article/pii/S0277953613000312?casa_token=L6R5_4yTgmwAAAAA:e5fAKRX8e11vbm1eI98HL4FHutzLfOCfIh_XL-F6XxIt32GOCICPzs-LXy6zhTkqTnLS-FwIZw

https://www.thelancet.com/journals/lancet/article/PIIS0140-6736(09)60098-2/fulltext

https://www.sciencedirect.com/science/article/pii/S0167629603000596?casa_token=dFiUW-aL8SIAAAAA:x0qGXKlf5mci44lkQGHTb3Lp5jPQsSNTrNyG6655J7m96HIc9vbzjtFpiJuI5xEaXdv98a4NaQ

Generally the paper seem to make good use of operational reports (WHO reports specially), but is limited when it comes to literature review. More lit review will enrich the paper.

- Finally, the language and style should be adjusted to ensure it meets the standards for academic British English.

Reviewer #2: The manuscript is well-written and organized. It aims to analyze the overall impact of prosperity on the mortality of selected NCDs to capture the differences between European countries using an appropriate analytical tool (joinpoint regression). However, some concerns should be taken into consideration:

*The title should include mortality, not health.

* There are no policy implications in the abstract.

* Lines (51-59) didn't include any references.

* In text citations need revision according to the journal requirements.

*The paragraph begins with line 201 needs more explanation for the method.

*Please add asterisks in table 3 to show the significance of coefficients.

* in line 341, please change "determinates" to "determinants".

* You need to add titles of x and y axis in Figure 4.

* Please mention the assumptions of jointpoint regression, and if it was tested before the implementation of the model.

* In line 92, add the reference of EUROstat database.

6. PLOS authors have the option to publish the peer review history of their article (what does this mean?). If published, this will include your full peer review and any attached files.

Reviewer #1: **Yes: **Abdulkarim Ekzayez

Reviewer #2: **Yes: **Sarah Ibrahim

---

## [Author Response · Author response to Decision Letter 0]

11 Oct 2023

Dear Editor,

Thank you for giving us the opportunity to revise the paper. Appreciating the valuable comments of the referees’, we hope that we were able to improve our manuscript and address all the concerns thoroughly. We genuinely believe this helped us to improve our paper. We hope that the manuscript is now suitable for publication in PLOS ONE.

We would like to go through the reviewers’ and editorial comments regarding the manuscript briefly. 

5. Review Comments to the Author

Reviewer #1:

- Intro: “This advantage predominantly stems from better education of the general population, higher quality of healthcare specialists, more advanced medical technology financed through general taxes and common social or private medical insurance”. This statement needs to be backed by evidence/reference.

 This statement has been backed by the references [10], [19], [20].

- Intro: “Overall, a higher level of wealth affects the individual and the population's health in two opposite ways”. Again, more evidence/ref is required here on how these assumptions/suggestions are made.

 This statement has been backed by the references [15], [16], [21].

- Results and discussions: the findings are well presented. One area of development could be to enrich the discussion further with contextualising external determinants of health disparities and providing some causal explanations. For example, the authors state that some types of neoplasms have lower or decreasing mortality rates in countries with higher Human Development Index, however further analysis of this connection would add further insight into the methodology and results.

 The discussion has been elaborated and additional context has been provided and backed by the references [41], [42], [43], [44], [452].

- Conclusion: the whole section lack any comparison with similar literature to situate the paper’s finding in the literature. Some of the papers that came to mind are:

https://www.sciencedirect.com/science/article/pii/S0277953613000312?casa_token=L6R5_4yTgmwAAAAA:e5fAKRX8e11vbm1eI98HL4FHutzLfOCfIh_XL-F6XxIt32GOCICPzs-LXy6zhTkqTnLS-FwIZw

https://www.thelancet.com/journals/lancet/article/PIIS0140-6736(09)60098-2/fulltext

https://www.sciencedirect.com/science/article/pii/S0167629603000596?casa_token=dFiUW-aL8SIAAAAA:x0qGXKlf5mci44lkQGHTb3Lp5jPQsSNTrNyG6655J7m96HIc9vbzjtFpiJuI5xEaXdv98a4NaQ

 A comparison with similar literature has been done - [19], [45], [60], [61], [62], [63], [64], [65]. 

Generally the paper seem to make good use of operational reports (WHO reports specially), but is limited when it comes to literature review. More lit review will enrich the paper.

 The literature review has been added in all sections.

- Finally, the language and style should be adjusted to ensure it meets the standards for academic British English.

 Professional proofreading has been performed on the revised version of the manuscript by The Cambridge Proofreading & Editing LLC.

Reviewer #2:

*The title should include mortality, not health.

- The title has been changed and includes ‘mortality’, i.e. “The Net Effect of Wealth on Health—Investigating Noncommunicable Diseases Mortality in the Context of Regional Affluence”. 

* There are no policy implications in the abstract.

- The policy implications have been added in the abstract.

* Lines (51-59) didn't include any references.

- Additional information with appropriate references has been added: [3], [7], [9], [11], [17], [18].

* In text citations need revision according to the journal requirements.

- The citations have been corrected according to the journal requirements.

*The paragraph begins with line 201 needs more explanation for the method.

 The method has been explained in detail: 

“Linear functions f_1 and f_2 obtained by the joinpoint regression method: 

〖MR〗_i^d (GDP_i;α_0,α_1,α_2,〖GDP〗^* )={■(f_1 (GDP_i )+ϵ_i,for i∈R¬_1 @f_2 (GDP_i )+ϵ_i,for i∈R¬_2 )┤ 

can be extrapolated to the regions with GDP outside their respective domains: R1 and R2. That is, the theoretical extension of the function f_2 covers now regions with lower GDP (R1), where theoretical extension of the function f_1 is applied to the regions with higher GDP (R2). Operating on those theoretical functions' extrapolations and the empirical values, by calculating the difference between them, we can acquire two net effect values: the net effect of poverty (𝑁𝐸_𝑃) and of affluence (NE_A).” 

*Please add asterisks in table 3 to show the significance of coefficients.

- The significance asterisks in Tab. 3, as well as Tab. 2 have been added.

* in line 341, please change "determinates" to "determinants".

- Done.

* You need to add titles of x and y axis in Figure 4.

- The titles of x and y axis has been added in Fig. 4.

* Please mention the assumptions of jointpoint regression, and if it was tested before the implementation of the model.

- Appropriate assumptions have been tested, and a proper statement has been added:

“We assume that the error terms (ϵ_i) are randomly distributed, independent with zero mean and a common bound for variances σ^2 for both subsets. This model can be estimated with an adjusted ordinary least squares method, by following the general theory of Pötscher and Prucha [39]. The resulting M-estimator turns out to be consistent under the same assumptions as standard OLS for the regressors and with the additional requirement of the corresponding parameter space being bounded.”

* In line 92, add the reference of EUROstat database.

- The reference of Eurostat database has been added – [22]

-------------

Journal Requirements:

- Corrected according to PLOS ONE's style requirements.

- The data file has been provided.

3. We note that Figures 1-3 and 5 in your submission contain [map/satellite] images which may be copyrighted. All PLOS content is published under the Creative Commons Attribution License (CC BY 4.0), which means that the manuscript, images, and Supporting Information files will be freely available online, and any third party is permitted to access, download, copy, distribute, and use these materials in any way, even commercially, with proper attribution. 

- All the maps in the manuscript are of our own compilation using Eurostat shapefiles of administrative borders (under © EuroGeographics for the administrative boundaries). Fig. 1-3 have been created in GeoDa software (under GNU Affero General Public License version 3) and Fig. 5 in ArcMap (under Education Institutional Small International Agreement version 11.1) The software and shapefiles are granted on the condition that the data will not be used for commercial purposes and that the source will be acknowledged – provided in the figures’ description.

4. Please review your reference list to ensure that it is complete and correct.

 We have reviewed the reference list to ensure it is complete and correct.

---

## [Decision Letter · Decision Letter 1]

23 Oct 2023

The net effect of wealth on health—investigating noncommunicable diseases mortality in the context of regional affluence

PONE-D-23-17022R1

Dear Dr. Żółtaszek,

We’re pleased to inform you that your manuscript has been judged scientifically suitable for publication and will be formally accepted for publication once it meets all outstanding technical requirements.

Kind regards,

Valentina Diana Rusu, PhD

Academic Editor

PLOS ONE

Additional Editor Comments (optional):

Reviewers' comments:

Reviewer's Responses to Questions

**Comments to the Author**

1. If the authors have adequately addressed your comments raised in a previous round of review and you feel that this manuscript is now acceptable for publication, you may indicate that here to bypass the “Comments to the Author” section, enter your conflict of interest statement in the “Confidential to Editor” section, and submit your "Accept" recommendation.

Reviewer #1: All comments have been addressed

Reviewer #2: All comments have been addressed

2. Is the manuscript technically sound, and do the data support the conclusions?

Reviewer #1: Partly

Reviewer #2: Yes

3. Has the statistical analysis been performed appropriately and rigorously? 

Reviewer #1: Yes

Reviewer #2: Yes

4. Have the authors made all data underlying the findings in their manuscript fully available?

Reviewer #1: Yes

Reviewer #2: Yes

5. Is the manuscript presented in an intelligible fashion and written in standard English?

Reviewer #1: Yes

Reviewer #2: Yes

6. Review Comments to the Author

Reviewer #1: Thanks for addressing all of the comments adequately. I think the paper is now ready for publication.

Reviewer #2: No more comments are needed, all the previous comments have been addressed by the author, and the manuscript is ready for publication

7. PLOS authors have the option to publish the peer review history of their article (what does this mean?). If published, this will include your full peer review and any attached files.

Reviewer #1: **Yes: **Abdulkarim Ekzayez

Reviewer #2: **Yes: **Sarah Assem Ibrahim

---

## [Editor Report · Acceptance letter]

27 Oct 2023

PONE-D-23-17022R1 

The net effect of wealth on health—investigating noncommunicable diseases mortality in the context of regional affluence 

Dear Dr. Żółtaszek:

I'm pleased to inform you that your manuscript has been deemed suitable for publication in PLOS ONE. Congratulations! Your manuscript is now with our production department. 

Kind regards, 

on behalf of

Dr. Valentina Diana Rusu 

Academic Editor

PLOS ONE